# The Role of CD36 in Cancer Progression and Its Value as a Therapeutic Target

**DOI:** 10.3390/cells12121605

**Published:** 2023-06-11

**Authors:** William W. Feng, Hannah T. Zuppe, Manabu Kurokawa

**Affiliations:** 1Lowe Center for Thoracic Oncology, Dana-Farber Cancer Institute, Boston, MA 02215, USA; 2Department of Medical Oncology, Dana-Farber Cancer Institute, Boston, MA 02215, USA; 3Department of Medicine, Harvard Medical School, Boston, MA 02215, USA; 4School of Biomedical Sciences, Kent State University, Kent, OH 44240, USA; 5Department of Biological Sciences, Kent State University, Kent, OH 44240, USA

**Keywords:** tumor microenvironment, lipid metabolism, cancer stem cells, metastasis, drug resistance, immune evasion, targeted therapy

## Abstract

Cluster of differentiation 36 (CD36) is a cell surface scavenger receptor that plays critical roles in many different types of cancer, notably breast, brain, and ovarian cancers. While it is arguably most well-known for its fatty acid uptake functions, it is also involved in regulating cellular adhesion, immune response, and apoptosis depending on the cellular and environmental contexts. Here, we discuss the multifaceted role of CD36 in cancer biology, such as its role in mediating metastasis, drug resistance, and immune evasion to showcase its potential as a therapeutic target. We will also review existing approaches to targeting CD36 in pre-clinical studies, as well as discuss the only CD36-targeting drug to advance to late-stage clinical trials, VT1021. Given the roles of CD36 in the etiology of metabolic disorders, such as atherosclerosis, diabetes, and non-alcoholic fatty liver disease, the clinical implications of CD36-targeted therapy are wide-reaching, even beyond cancer.

## 1. Introduction

CD36 is a multifunctional cell surface protein belonging to the scavenger receptor class B family [1]. It was originally identified as a major platelet membrane glycoprotein, platelet glycoprotein IV (GPIV) [2,3]. Soon after its discovery, GPIV was shown to be identical to the leukocyte differentiation antigen, CD36 [4], and was demonstrated to be a receptor for thrombospondin-1 (TSP-1), a glycoprotein involved in platelet adhesion and an endogenous inhibitor of angiogenesis [5,6,7]. In 1993, CD36 was found to function as a receptor for oxidized low-density lipoprotein (oxLDL) in macrophages and a fatty acid (FA) transporter in adipocytes, thereby establishing its role as a scavenger receptor [8,9].

Although it is most often recognized for its role as a FA transporter, CD36 also serves as a receptor that is involved in the transduction of several signaling pathways (Figure 1). Upon stimulation, CD36 has been demonstrated to mediate the activation of SRC family non-receptor tyrosine kinases (Figure 1A), such as FYN and LYN, and serine/threonine kinases of the MAPK pathway, such as p38 and JNK [10,11,12]. The effectors of these kinases have not been fully elucidated, but several studies have identified that the cell adhesion kinases, PYK2 and FAK, play important roles downstream of CD36 [10,11,12]. However, it is important to note that CD36 signaling is highly cell-type specific. For example, LYN and JNK2 are primarily activated in macrophages and platelets in response to stimuli, while FYN and p38 MAPK are preferentially involved in endothelial cell responses [10]. The factor(s) determining the associated downstream signaling proteins and functional consequences remains to be fully elucidated.

In recent years, CD36 has quickly emerged as an attractive therapeutic target in cancer due to its multifaceted role in tumor biology. In addition to functioning as a FA transporter in cancer cells, CD36 also acts as a conduit for crosstalk with other cell types of the tumor microenvironment (TME), namely adipocytes, which supply FAs for cancer cell uptake [13,14,15]. This review will summarize the role of CD36 in promoting the growth and stemness of cancer cells, providing energetic support for metastasis, mediating the acquisition of drug resistance to a wide range of therapeutic agents, and facilitating immune suppression in the TME. We will also highlight a few of the current antibodies and small molecule inhibitors that target CD36, including VT1021, the only CD36-targeting drug currently in clinical trials.

**Figure 1 cells-12-01605-f001:**
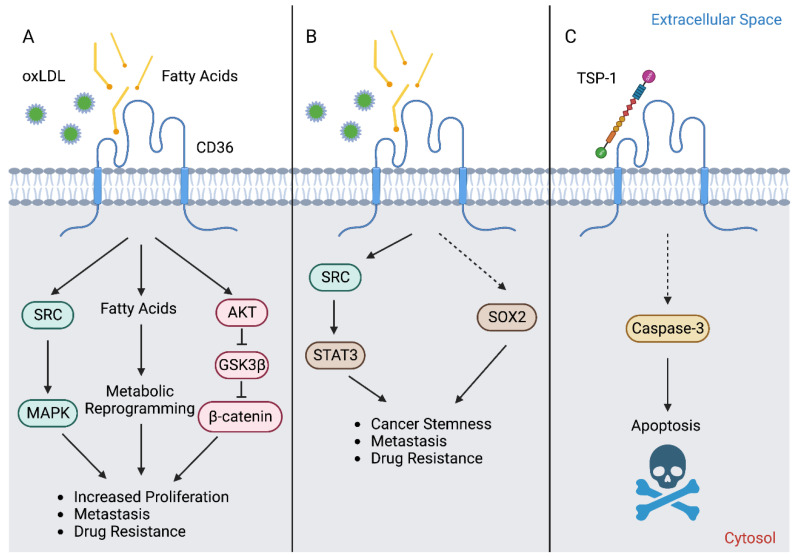
CD36 exhibits multiple distinct functions in cancer cells. (**A**) The primary mechanism of action of CD36 is its activity as a FA transporter, which supplies the cell substrates for bioenergetic purposes to promote proliferation, metastasis, and drug resistance. CD36-mediated FA uptake has also been demonstrated to facilitate activation of the SRC/MAPK, as well as AKT/GSK3β/β-catenin signaling axes, to support these processes. (**B**) Moreover, CD36-mediated activation of STAT3 or SOX2 has been shown to induce epithelial–mesenchymal transition (EMT) and promote cancer stemness, metastasis, as well as drug resistance. The activation of STAT3 occurs following FA or oxLDL binding to CD36, presumably via the recruitment of SRC [16,17]. It was shown that CD36 could physically interact with STAT3 in breast cancer cells co-cultured with adipocytes [17]. Importantly, this interaction could be abolished with the CD36 inhibitor sulfo-N-succinimidyl oleate (SSO), indicating that the binding of a ligand (FA or oxLDL) to CD36 is required for the interaction between CD36 and STAT3 [17]. Similar results were observed in glioblastoma stem cells, where 2-methylthio-1,4-naphthoquinone (MTN), a pharmacologic inhibitor of CD36 that blocks oxLDL uptake, reduced ATK and STAT3 activation in addition to reducing total SOX2 levels [16]. Although not directly assessed in these studies, it is speculated that the activation of STAT3 was a result of SRC family kinase recruitment to CD36 [17]. The ligand-mediated activation of CD36 is thought to promote the transcription of SOX2 [18]. (**C**) Although it is canonically responsible for anti-angiogenic activity in endothelial cells, the TSP-1-CD36 apoptotic signaling cascade was recently demonstrated to remain intact in cancer cells. This discovery spurred the development of VT1021, a first-in-class TSP-1-inducing compound that elicits CD36-mediated cell death and is currently being investigated in clinical trials. The role of CD36-mediated signal transduction in cancer cells remains poorly understood and requires further study.

## 2. Regulation of CD36 Expression and Activity

The *CD36* mRNA codes for a 472-amino-acid-long protein. However, the mature CD36 protein is 471 amino acids long due to loss of the initiator methionine [6]. The majority of the CD36 protein resides in an extracellular region (Gly30-Asn439), anchored to the plasma membrane by two transmembrane domains at the N- and C-terminal ends (Gly8-Val29 and Leu440-Ile461) (Figure 2). The extracellular domain contains two large hydrophobic binding pockets, which are essential for FA and oxLDL uptake activity (Figure 2). Although only 17 amino acids of CD36 protein are cytoplasmic (Gly2-Cys7 and Ser462-Lys472), these residues are critical for CD36 maturation and signaling activity. As detailed below, CD36 is a heavily post-translationally modified protein.

### 2.1. Glycosylation

Although the predicted size of the CD36 protein is 53 kDa based on its amino acid composition, the observed molecular weight across tissues is often ~88 kDa due to glycosylation. N-linked glycosylation is a critical post-translational modification required for the maturation, stabilization, and proper trafficking of most cell surface proteins. CD36 harbors 10 potential N-linked glycosylation sites, which are all located in its extracellular domain (Figure 2). A mutagenesis study has demonstrated that when all 10 asparagine residues are mutated to glutamines, CD36 is retained in the cytoplasm and is unable to localize to the cell surface [19]. Notably, no single residue alone is solely responsible for determining the trafficking fate of CD36. In fact, the glycosylation pattern required for cell surface localization seems to be flexible due to a degree of functional redundancy in these residues. For instance, the restoration of either the first seven glycosylation sites (Asn79, Asn102, Asn134, Asn163, Asn205, Asn220, and Asn235) or the last three glycosylation sites alone (Asn247, Asn321, and Asn417) was sufficient to partially rescue CD36 cell surface localization [19]. However, the greatest degree of rescue could be seen when all 10 glycosylation sites were restored [19]. Importantly, N-linked glycosylation of CD36 does not appear to affect ligand binding [19].

O-linked-N-acetylglucosaminylation (O-GlcNAcylation) is another form of glycosylation that involves the addition of the sugar moiety N-acetylglucosamine (GlcNAc) with serine or threonine residues of proteins. It was recently reported that CD36 harbors two potential O-GlcNAcylation sites on Ser468 and Thr470 [20]. O-GlcNAc Transferase (OGT) and O-GlcNAcase (OGA) are the endogenous enzymes that catalyze the addition and removal of O-GlcNAc from proteins, respectively. The pharmacologic inhibition of OGA was recently shown to activate the NF-κB pathway in gastric cancer cells to markedly induce CD36 transcription, as well as increase the O-GlcNAcylation of CD36 protein [20]. The O-GlcNAcylation of CD36 subsequently enhanced cellular FA uptake and promoted metastasis of gastric cancer models [20]. Interestingly, the high expression level of both CD36 and OGT predicted a worse clinical outcome in gastric cancer patients [20]. Although the O-GlcNAcylation of CD36 has also been reported in cardiomyocytes [21], it remains unclear how prominent role O-GlcNAcylation plays to regulate CD36 functions in other cell types and cancers.

### 2.2. Ubiquitination

Lys469 and Lys472 are the only lysine residues within the CD36 intracellular domain, and both have been previously identified as critical poly-ubiquitination sites that target CD36 protein for proteasomal degradation [22] (Figure 2). In myoblast cells, FA binding induces CD36 poly-ubiquitination, thereby destabilizing the protein and reducing cellular FA uptake [22]. Interestingly, insulin exhibits the opposite effects in this model and stabilized CD36 to further enhance FA uptake [22]. However, the relevance of these studies in the setting of cancer is unclear since dietary FA supplementation has been shown to increase the CD36 expression level, which then promotes tumor growth and metastasis in xenograft models of oral squamous cell carcinoma (OSCC) [23], cervical [7], and gastric cancers [20]. Moreover, co-culture studies have revealed that omental adipocytes can induce the expression of CD36 in ovarian cancer cells to facilitate the uptake of adipocyte-derived FAs into cancer cells, promoting tumor growth and enhancing metastasis [14]. Therefore, CD36 levels in cancer cells may not be primarily regulated via FA-induced degradation.

At present, Parkin is the predominant E3 ubiquitin ligase known to promote the degradation of the CD36 protein. In mice, hepatic levels of Parkin were demonstrated to be elevated by a high-fat diet (HFD), which resulted in stabilizing CD36 protein via mono-ubiquitination [24]. However, a recent study reported that Parkin knockout mice exhibited no difference in CD36 protein levels in the brain [25]. On the other hand, it was demonstrated that the deletion of the master regulator of energy homeostasis, AMPK, promoted Parkin-mediated poly-ubiquitination and the degradation of CD36 in intestinal epithelial cells [26]. Therefore, the role of the Parkin-mediated regulation of CD36 appears to be largely context- and tissue-specific. The action of E3 ubiquitin ligases is counteracted by deubiquitinating enzymes, deubiquitinases, which stabilize the substrate proteins by removing ubiquitin. Three deubiquitinases are known to stabilize CD36 protein in macrophages: USP10 [27], USP14 [28], and UCHL1 [29]. All in all, whether Parkin, USP10, USP14, or UCHL1 plays a role in the regulation of CD36 protein stability in cancer cells remains poorly understood.

### 2.3. Palmitoylation

CD36 contains four intracellular cysteine residues, which are all known palmitoylation sites (Cys3, Cys7, Cys464, and Cys466) [30] (Figure 2). Interestingly, the maturation of palmitoylation-deficient CD36 mutant protein was markedly slower as compared to that of the wildtype, CD36 [31]. Additionally, the mutant CD36 also exhibited a shorter protein half-life, as well as a defect in incorporation into lipid rafts, which impairs its ability to import oxLDL [31]. Strikingly, another study has also demonstrated that the loss of each of the four palmitoylation sites is sufficient to abrogate insulin or the AMPK-activation-induced cell surface localization of CD36, suggesting that all four palmitoylation sites are critical for this process [32]. Recently, the palmitoyl acyltransferases, DHHC4 and DHHC5, were identified to play indispensable roles in CD36 maturation [33]. DHHC4 palmitoylates synthesized CD36 in the Golgi, and the silencing of DHHC4 expression alone blocked nearly all CD36 cell surface localization. In addition, DHHC5 functions at the cell surface and helps stabilize CD36 by preventing depalmitoylation [33]. Lastly, two recent studies have independently reported that CD36 acylation/deacylation and glycosylation/deglycosylation are critical steps that govern both the uptake of extracellular FAs, as well as the trafficking of CD36 to/from lipid droplets during lipolysis in adipocytes [15,34]. Therefore, the palmitoylation of CD36 plays a critical role in its protein functions as an FA/oxLDL transporter.

### 2.4. Other Post-Translational Modifications

While glycosylation and palmitoylation are critical for CD36 maturation, phosphorylation plays a role in the regulation of ligand binding and FA uptake, as well as signal transduction activity. CD36 has two known phosphorylation sites in its extracellular domain, Thr92 and Ser237, which are phosphorylated using PKC and PKA, respectively [35,36] (Figure 2). It was shown that Thr92 phosphorylation using PKC can take place intracellularly before trafficking to the cell surface [37]. Under basal conditions in melanoma cells, CD36 is predominantly non-phosphorylated and presumed to be active [37]. The treatment of these cells with a PKC activator resulted in an increase in the rates of CD36 phosphorylation and the concomitant reduction of TSP-1 peptide binding, which consequently abolished the ligand-induced recruitment of SRC to CD36 [37]. Likewise, it was shown that treatment with a PKA inhibitor peptide reduced the rate of palmitate uptake into the platelets [38]. In contrast to Thr92 phosphorylation using PKC, however, the role and the mechanism of Ser237 phosphorylation using PKA remain poorly characterized to date. It has been shown that PKA can function at the cell surface and phosphorylate the extracellular domain of CD36 [36]. However, this does not rule out the possibility of the intracellular phosphorylation of CD36 using PKA prior to trafficking to the plasma membrane. Importantly, an earlier study reported that PKA could phosphorylate CD36, but less efficiently than PKC [35].

The acetylation of human CD36 has been described for Lys52, Lys16, Lys231, and Lys403 [39] (Figure 2). The acetylation of Lys166 may impact the FA uptake activity of CD36 since acetylated Lys166 and FA-bound Lys164 have not been found on the same protein in mass spectrometry experiments [39]. However, the functional significance and prevalence of CD36 acetylation in cancer remain uncharacterized.

**Figure 2 cells-12-01605-f002:**
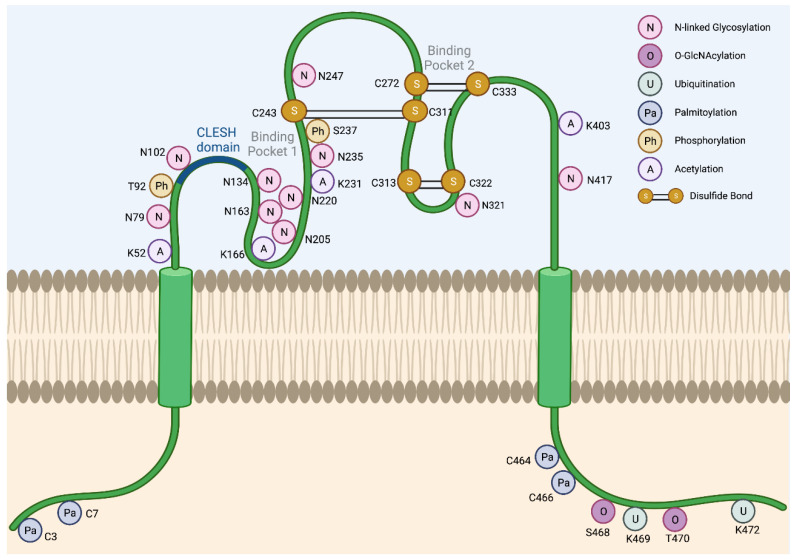
Post-translational modifications of CD36. Several post-translational modifications have been described for CD36 including N-linked glycosylation (N), O-GlcNAcylation (O), ubiquitination (U), palmitoylation (Pa), phosphorylation (Ph), acetylation (A), and disulfide bonds (S-S). The CLESH domain (93–120) of CD36 plays a critical role in the interaction with thrombospondins. Lys164 is known to be the binding site of SSO, FAs, and oxLDL [39].

The extracellular domain of CD36 harbors three disulfide bonds that are critical for its structure (Figure 2). Interestingly, the Cys272-Cys333 disulfide bond in CD36 was shown to be cleavable using hydrogen sulfide (H_2_S), and the loss of this specific motif enhanced FA uptake and metastasis of gastric cancer cells [40]. The translatability of this finding to other cancer types and models requires further study.

### 2.5. Transcriptional Regulation of CD36

In addition to the regulation of CD36 maturation, trafficking, activity, and stability via post-translational modifications, CD36 is also regulated at the transcriptional level. Many transcription factors have been shown to regulate CD36 expression, including PPARγ [41,42,43], LXR [42], PXR [42], C/EBPα [44], STAT3 [17,45], HIF1α [14,20], SOX2 [20], and NF-κB [14,20]. Some of these transcription factors are known to play a role in human diseases, including cancer. For instance, the activation of NF-κB could trigger lipid metabolic rewiring in ovarian cancer cells, thereby activating a transcriptional program that leads to increased CD36 expression, which in turn can be critical for ovarian cancer survival and metastasis [14]. Interestingly, Protein Kinase D1 (PKD1) is known to be a potent activator of NF-κB [46,47], and it is often aberrantly activated in cancer [48,49]. PKD1 signaling has been demonstrated to mediate the induction of CD36 to facilitate the maintenance of a stem-like phenotype in pancreatic neuroendocrine tumors [49]. As described below, the transcriptional upregulation of CD36 has been well documented in other cancers. However, transcription factors responsible for CD36 upregulation in cancer are largely unknown.

## 3. CD36 in Cancer

Accumulating evidence strongly suggests that CD36 plays critical roles in many different cancer types (Table 1). CD36 is widely expressed in cancers of the brain [16], breasts [50,51,52,53], colon [53], gastric [20,40,54,55], mouth [23], ovaries [14,56], and pancreas [57], among many others [50,58,59,60,61,62]. In cancer cells, CD36 functions primarily as a FA transporter that mediates the uptake of extracellular FAs to support the metabolic needs of the cell, such as fueling the energetic demands of metastasis [14,23,54] (Figure 1A). Data from the TCGA database [63] also indicates that CD36 is amplified and mutated in a variety of cancer types (Figure 3). However, mutations in CD36 are rare, and according to the ClinVar Miner database, CD36 mutations are not pathogenic in cancer (https://clinvarminer.genetics.utah.edu/variants-by-gene/CD36, accessed on 20 February 2023). Additionally, according to the catalog of the Somatic Mutations in Cancer (COSMIC) database, CD36 mutations have not been reported to be linked to drug resistance in cancer (https://cancer.sanger.ac.uk/cosmic/gene/analysis?ln=CD36, accessed on 20 February 2023). Therefore, the deleterious effects of CD36 in cancer are likely due to increased expression and activity rather than the gaining of function mutations. In addition to CD36, other proteins involved in FA uptake and synthesis, such as fatty acid synthase (FASN), are also frequently amplified and/or overexpressed in metastatic tumors [64]. Several studies have shown that the inhibition of FASN can lead to the compensatory upregulation of CD36 expression and that inhibiting both pathways is lethal to cancer cells, highlighting the essentiality of FA pool maintenance in cancer [53,62,65]. It was also recently described that CD36 plays a central role in cancer cell metastasis and that genetically or pharmacologically inhibiting CD36 can perturb tumor growth and metastasis across several cancer types [14,16,23,54,61]_._ Importantly, work from our group has also demonstrated a role for CD36 in mediating the acquisition of resistance to HER2-targeted therapies in breast cancer [52]. We found that CD36 expression was elicited following HER2-targeted therapy in human breast tumors and demonstrated that high CD36 levels in these tumors were significantly associated with worse clinical outcome in patients [52]. Importantly, a high CD36 expression level has also been shown to correlate with worse prognosis in the brain [16], cervical [60], colorectal [53], gastric [20,54,55], pancreatic [57], and prostate [62] cancers.

## 4. CD36 in Cancer Stemness and EMT

Multiple studies have shown that CD36 is enriched in cancer stem cell (CSC) populations from several cancer types, and its expression is often associated with an enhanced metastatic capacity [16,23,51]. In glioblastoma, for example, CD36 was found to be a key marker of stemness and was implicated in facilitating tumor initiation and growth [16]. It was shown that CD36 levels decreased over the course of the cellular differentiation of CSCs [4716 Conversely, the downregulation of CD36 via an siRNA treatment was shown to result in loss of self-renewal and tumor initiating ability [16]. Studies have also shown that CD36 expression is highly associated with EMT gene signatures across multiple cancer types [17,59,60,64]. In brain, breast, and liver cancers, an elevated CD36 expression level leads to enhanced FA uptake, which is shown to promote migration, invasion, and EMT via the activation of STAT3 signaling [16,17,59] (Figure 1B). Importantly, this EMT phenotype could be reversed with the pharmacologic inhibition [16,17] or genetic ablation of *CD36* [17,59]. Similarly, *CD36* RNAi was also able to reverse TGFβ-mediated EMT in cervical cancer [60]. These results suggest that CD36 serves as a potentially useful biomarker of CSC and EMT. Since CSCs are subpopulations of cancer cells that exhibit a self-renewal capability and resistance to chemotherapies, targeting CD36 may offer a strategy to target these notoriously difficult-to-eliminate populations responsible for therapy relapses among patients.

Interaction with stromal populations in the tumor microenvironment (TME) may be an important factor in modulating CD36 and CSC/EMT marker expression in cancer cells. It was demonstrated that co-culture with adipocytes was sufficient to enhance the expression of CD36 and EMT transcription factors (TWIST1, SNAIL, and ZEB1), induce the expression of mesenchymal markers (N-cadherin, MMP9, and Vimentin), and decrease the expression level of E-cadherin in breast cancer cells [17]. Many of these trends could be further enhanced via CD36 overexpression in cancer cells [17]. Moreover, adipocyte-co-cultured cancer cells displayed an increased expression level of stem cell markers (CD44, OCT4, and SOX2) which could also be further promoted via CD36 overexpression [17] (Figure 1B). Another recent study demonstrated that the co-culturing of prostate cancer cells with primary cancer associated fibroblasts (CAFs) induced over a 20-fold increase in the expression level of SOX2 in cancer cells [18]. Moreover, SOX2 expression in prostate cancer cells could also be further induced via palmitate supplementation to the in vitro co-culture system or via HFD in an in vivo co-xenograft mouse model [18]. Notably, this elevated SOX2 expression could be inhibited by a function-blocking anti−CD36 antibody [18]. These studies demonstrate that the role of CD36 in regulating cancer stemness may involve crosstalk with other cell types in the TME, adding yet another layer of complexity to its profile in tumor biology.

## 5. CD36 in Metastasis

Several recent studies have implicated CD36 in facilitating cancer cell metastasis in a wide variety of cancer types [14,16,23,60,61]. Namely, a seminal study conducted by Benitah and colleagues demonstrated that cells with a high CD36 expression level were significantly enriched in OSCC stem cell populations [23]. These cells exhibited a strong lipid metabolism gene signature and exhibited improved metastatic and tumor-initiating capacities [23]. These phenotypes appear to be regulated via CD36-mediated FA uptake since xenograft tumor growth and metastasis are enhanced by HFD and could be attenuated by CD36 function blocking antibodies [23]. Similar findings were reported in gastric cancer, where HFD or CD36 overexpression was shown to enhance the peritoneal dissemination of gastric cancer cells via the AKT-mediated inactivation of GSK3β and the subsequent stabilization of β-catenin [54] (Figure 1A). As described earlier, other studies found that omental adipocytes had the ability to induce the expression of CD36 in neighboring ovarian cancer cells to facilitate the uptake of adipocyte-released FAs [14,23]. A similar form of crosstalk has also been described to occur between adipocytes and breast cancer cells [13,15,17]. Notably, targeting CD36 with function-blocking antibodies or genetic ablation reduced the tumor burden and metastasis in OSCC, gastric, and ovarian cancer xenograft mouse models, with reductions of up to 90% observed in the size of lymph node metastases [14,23,54]. 

Interestingly, it was shown that omental adipocytes also induced the expression of another FA transporter, FABP4, in ovarian cancer cells to facilitate their metastasis [76]. Furthermore, FASN-mediated FA synthesis has recently been shown to be critical for fueling brain metastasis in breast cancer [77]. Lastly, leukemia stem cells exhibiting a high CD36 expression level and elevated rates of FA oxidation (FAO) have been shown to preferentially home to gonadal adipose tissue, indicating a preference for lipid-rich microenvironments that facilitates their FA needs [58]. Together, these studies highlight that FA metabolism appears to be an essential fuel source for metastasis, although cancer cells could evolve a variety of different mechanisms, in both CD36-dependent and -independent manners, to meet their metabolic demands. 

## 6. CD36 in Cancer Drug Resistance

CD36 has recently been implicated in the development of resistance to several distinct classes of therapies in cancer. For instance, leukemia stem cells with a high CD36 expression level have been shown to be resistant to cytarabine, doxorubicin, etoposide, SN38, and irinotecan [58]. This resistant population was also shown to evade drug treatment via forming a niche in gonadal adipose tissue of leukemic mice, where they utilize CD36-mediated FA uptake and elevated FAO levels to survive [58]. Importantly, the genetic ablation of CD36 in leukemia cells abrogated gonadal adipose homing and conferred sensitization to chemotherapy [58]. 

Our group previously demonstrated that CD36 mediates the acquisition of resistance to the tyrosine kinase inhibitor lapatinib in HER2^+^ breast cancer [52]. We identified that the upregulation of CD36 in resistant cells confers metabolic plasticity and allows them to survive challenging environmental conditions, such as nutrient deprivation and drug treatment [52]. Notably, we found that CD36 expression predicts a poor prognosis among patients with HER2^+^ breast cancer and that its expression also increases among patients following a treatment with HER2-targeted therapy [52]. Interestingly, CD36 was also found to be elevated in tamoxifen-resistant MCF7 cells, which is a HER2^-^ breast cancer cell line [41]. Moreover, CD36 gene silencing with siRNA was able to re-sensitize resistant MCF7 cells to tamoxifen [41]. This study suggests that CD36-mediated therapy resistance may not be specific to the HER2^+^ subtype of breast cancers.

In BRAF-mutant melanoma, CD36 was found to be the most consistently upregulated cell surface marker following MAPK inhibition [72]. However, CD36 does not appear to function as a FA transporter in this setting since the rates of FA uptake and FAO were both unaffected by CRISPR-mediated CD36 knockout [72]. While the role of CD36 in this setting remains unclear, the level of CD36 cell surface expression remained stable in MAPK inhibitor-resistant cells and offers the possibility to serve as a potential biomarker for this drug resistant population in melanoma. 

Another recent study demonstrated that the FAK-YAP1 signaling axis appears to play a key role in the establishment of drug-tolerant persister cells that mediate resistance to EGFR-, ALK-, and KRAS-targeted therapies in lung cancer [78]. While it has not been explicitly investigated, it is tempting to speculate that CD36 could be involved in this resistance mechanism since FAK serves as one of the major kinases through which CD36 facilitates signaling. Moreover, CD36-mediated oxLDL uptake has been previously linked to promoting the oncogenic activity of YAP in the etiology of hepatocellular carcinoma [79]. However, future studies are needed to formally test this hypothesis.

Apart from kinase inhibitors, CD36 has also been shown to be upregulated in gemcitabine-resistant pancreatic ductal adenocarcinoma (PDAC) cells, and it predicts a poor prognosis among PDAC patients [57]. A high CD36 expression level has also been described in cisplatin-resistant ovarian cancer cells and in leukemia stem cells exhibiting resistance to cytarabine and doxorubicin [58,74]. Importantly, genetic silencing or the ablation of *CD36* was shown to restore sensitivity to chemotherapy in these studies [57,58]. Moreover, bortezomib-resistant mantle cell lymphoma cells were also found to exhibit elevated CD36 expression levels, increased FA uptake, and could similarly be re-sensitized to bortezomib with a function-blocking antibody of CD36 [68]. Lastly, CD36 has also been shown to be involved in the resistance to imatinib in chronic myeloid leukemia in several studies [69,80]. Interestingly, one of these studies demonstrated that imatinib-resistant cells can secrete exosomes containing CD36, which could confer resistance to drug naïve cells, unveiling a novel mechanism of CD36-mediated drug resistance [80]. Together, these studies demonstrate that CD36 has the potential to facilitate the development of resistance for a large number of therapies across cancer types via a number of functionally distinct and context-dependent mechanisms. 

## 7. CD36 in Immune Evasion

CD36 is critical for the development and maturation of several immune populations in the TME. Wang et al. recently described an essential role for CD36 in the establishment and maintenance of immunosuppressive regulatory T cells (T_regs_) in melanoma [56]. Interestingly, while CD36 is dispensable in peripheral T_regs_, it is specifically required to support intratumoral T_reg_ suppressive functions, as CD36 helps to maintain T_reg_ mitochondrial fitness in order to survive the harsh TME [56]. Therefore, T_reg_-specific CD36 knockout or treatment with a function-blocking antibody suppressed tumor growth and promoted anti-tumor immunity, which was even more pronounced when it was combined with an immune checkpoint inhibitor, such as an anti-PD-1 antibody [56]. While CD36 expression promotes the survival of intratumoral T_regs_, two recent studies have shown that CD36-mediated oxLDL uptake could cause the dysfunction of tumor-infiltrating CD8^+^ T cells, including the suppression of cytotoxic cytokine production, via promoting ferroptosis as accumulated oxLDL induces lipid peroxidation in the cells [73,81].

In addition to T cells, CD36 is also highly expressed on pro-tumorigenic M2 macrophages, where it facilitates the uptake of oxLDL and the phagocytosis of apoptotic cells [66,71,82,83]. Moreover, a recent study demonstrated that the genetic ablation of *Cd36* was able to abrogate the development of M2 tumor-associated macrophages (TAMs) and preferentially promote the maturation of pro-inflammatory M1 macrophages [71]. This study also demonstrated that macrophage-specific CD36 knockout enhanced cytotoxic cytokine production in CD8^+^ T cells and attenuated liver metastasis, indicating a crucial role for CD36 in promoting immune evasion [71]. 

Colony stimulating factor receptor 1 (CSF1R) is a critical macrophage proliferation marker that is currently being actively investigated in clinical trials as a therapeutic target in cancer. CSF1R inhibition has been shown to deplete M2 macrophages from the TME and repolarize them into having an M1 phenotype [84]. Notably, CSF1R activation has been demonstrated to induce the expression of CD36 in bone marrow-derived macrophages, raising the interesting possibility that macrophage repolarization following CSF1R inhibition may occur secondary to attenuated CD36 activation [83]. Future studies into elucidating the role of CD36 as a potential therapeutic target in macrophage repolarization are warranted. Nevertheless, these studies demonstrate the multifaceted roles CD36 plays in establishing immunosuppressive TMEs and shed light on the potential for remodeling the tumor immune milieu with anti-CD36 therapies. 

## 8. CD36 as a Therapeutic Target

Given its broad expression pattern and eclectic roles in promoting the growth and treatment resistance of many different types of cancer, CD36 is emerging as a promising therapeutic target. Historically, most of the inhibitors developed against CD36 are small molecule inhibitors or FA mimetics (Table 2). SSO is a FA analog that inhibits long-chain FA and oxLDL uptake by CD36 via irreversibly binding Lys164, the FA binding site of CD36 [39]. Function-blocking antibodies aside, SSO is arguably the most common inhibitor used for the perturbation of CD36 activity in in vitro studies. The pharmacological inhibition of CD36 by SSO has also been used in several pre-clinical studies in mice and rats for the treatment of cancer and cardiovascular diseases. However, reports of the in vivo tolerability of SSO have been conflicting [14,53,85]. Unfortunately, this uncertainty in safety has resulted in doubt over its translatability as a viable therapeutic agent in humans.

A variety of other structurally distinct small molecules with anti-CD36 activity have also been isolated from natural sources and characterized in the past. Salvianolic acid (SAB) is a polyphenolic compound extracted from the roots of *Salvia miltiorrhiza*. It is a naturally occurring CD36 inhibitor that blocks the uptake of oxidized lipids in macrophages and has been demonstrated to normalize metabolic dysfunction in diet-induced obese mice [86,87]. MTN is another naturally occurring metabolite isolated from a sponge-associated bacterium [88]. MTN was shown to inhibit cancer cell growth in several studies and exhibit anti-angiogenic activity in chick chorioallantoic membrane assays [16,88]. Aside from natural products, a novel approach to targeting CD36 was also demonstrated via synthetically conjugating a lipid tail to a platinum chemotherapeutic payload to generate a cytotoxic FA mimetic [74]. Utilizing these FA-linked Pt (IV) prodrugs, it was demonstrated that increased CD36 expression in cisplatin-resistant ovarian cancer cells could be leveraged to overcome cisplatin resistance [74].

Another approach to targeting CD36 would be to elicit the TSP-1-CD36-mediated apoptotic signaling pathway [89,90,91], rather than modulating the lipid uptake functions of CD36 (Figure 1C). It was initially demonstrated that a peptide derived from the glycoprotein prosaposin can induce CD36-mediated apoptosis in ovarian cancer via promoting TSP-1 expression in bone marrow-derived cells [75]. This TSP-1-CD36 axis is canonically responsible for anti-angiogenic functions in endothelial cells. However, the TSP-1-CD36 axis apparently remains intact in cancer cells [75]. This prosaposin-derived peptide has since been further developed and is currently being evaluated in clinical trials as VT1021, the only CD36-targeting agent in trials for cancer therapy at this time [92,93]. This first-in-class compound is a cyclic pentapeptide that has been demonstrated to reprogram myeloid-derived suppressor cells (MDSCs) to produce TSP-1 in the TME, which can then bind to CD36 and CD47 on tumor and endothelial cells [92,93]. This response ultimately results in the apoptosis of tumor and endothelial cells via a number of different mechanisms including the activation of CD36-mediated apoptotic signaling, blocking the CD47 “don’t eat me” immune checkpoint, promoting M2-to-M1 TAM repolarization, as well as increasing the rates of the activation and infiltration of cytotoxic lymphocytes (CTLs) [93]. Demonstrating favorable results in a Phase I/II trial in ovarian, pancreatic, triple-negative breast cancer, and glioblastoma (NCT03364400) [92], VT-1021 has currently advanced to Phase II/III for the treatment of glioblastoma (NCT03970447) [93]. It may be worth noting that VT1021 is not the first compound developed to induce CD36-mediated apoptosis in cancer cells. Previously, a TSP-1-mimetic, ABT-510, showed tolerability in cancer patients in Phase I studies both as a single agent [94] and in combination with chemotherapy and/or radiotherapy [95,96]. However, ABT-510 faltered in Phase II trials due to a lack of efficacy [97,98,99]. Fortunately, VT1021 has been reported to be well tolerated by patients and exhibits robust clinical activity even as a single agent across the tumor types tested [92,93]. 

The mechanism of CD36-mediated apoptosis has been best characterized to date in endothelial cells, where TSP-1 binding to CD36 directly activates a signaling cascade involving the FYN-p38-caspase-3 axis [89] (Figure 1C). In addition, TSP-1 was also demonstrated to mediate anti-angiogenic effects via concurrently inducing the transcription of Fas and FasL both in vitro and in vivo [100]. Although a few studies have reported that TSP-1-CD36-mediated apoptosis occurs in ovarian cancer [75] and leukemia [101]; this aspect of CD36 biology in cancer cells has remained largely unexplored and requires further study. 

In addition to small molecule CD36 inhibitors, many pre-clinical studies often employ function-blocking antibodies to specifically inhibit CD36 in vitro and in vivo (Table 2). As discussed earlier, several studies have demonstrated that these blocking antibodies can reduce tumor burden and metastasis either as single agents, or in combination with targeted therapies [14,23,54,56,73,81]. Moreover, the inhibition of CD36, either genetically or with function-blocking antibodies, was shown to be even more efficacious when it is combined with anti-PD-1 antibodies via further eliciting anti-tumor immunity in immunocompetent mice, thereby adding another layer of therapeutic value to targeting CD36 in cancer [56,81]. Unfortunately, because commonly employed CD36 function-blocking antibodies to date have been mouse-derived antibodies (IgG FA6.152 and IgA JC63.1), their translatability into human studies has been extremely limited by the possibility of human anti-mouse antibody (HAMA) immune responses. In this regard, Salvador Aznar Benitah, whose group first implicated CD36 in the metastasis of OSCC [23], recently founded ONA Therapeutics in Spain to work on developing a humanized CD36 antibody, which is expected to enter human trials in the near future [102]. There is a lot of enthusiasm in the scientific community about the potential of this endeavor, and we eagerly await the results of the study and the translational implications of the results for future CD36 targeting studies because, to date, the effects of CD36 inhibition in humans remain largely unknown. Considering that the major phenotype of global CD36 knockout mice and humans born with CD36 deficiency is hypertriglyceridemia [103,104], we are curious whether similar effects will also be seen in humans treated with anti-CD36 therapy. Additionally, CD36 deficiency in humans has also been associated with increased predisposition to developing insulin resistance [104,105]. Therefore, it will be critical to assess whether systemic anti-CD36 therapy results in the same side effect.

While there remains a lot of optimism about VT1021 and humanized CD36 antibodies in the context of cancer therapies, we anticipate that these molecules will also find broad utility in the treatment of other CD36-associated metabolic diseases. CD36 has been previously evaluated as a therapeutic target in diseases other than cancer, such as atherosclerosis, non-alcoholic fatty liver disease, and diabetes [85,87,88,106,107,108]. Notably, because many of these metabolic disorders often present as co-morbidities in cancer patients, it is feasible that anti-CD36 therapy could also have the potential to simultaneously treat those disorders, adding even further therapeutic value for this emerging class of inhibitors. For instance, AP5055 and AP5258 were previously identified from a high-throughput screening and were shown to reduce atherosclerosis and hypertriglyceridemia, as well as improve insulin sensitivity and glucose tolerance in rodent models [106]. Puerarin is a naturally occurring isoflavonoid from the roots of *Pueraria lobata* and has been demonstrated to reduce CD36 trafficking to the cell surface of myocytes to inhibit FA uptake and alleviate diabetic cardiac myopathy [107]. Lastly, EP80317 is a synthetic peptide ligand of CD36 that has been shown to mitigate the development of hypercholesterolemia and atherosclerosis in mice [109,110]. EP80317 also exhibits cardioprotective effects in murine models of myocardial ischemia and reperfusion injury, indicating that CD36 could also be a valuable target in ischemic cardiopathy [111]. All in all, CD36 inhibition appears to have the potential to confer numerous mechanistically distinct therapeutic benefits to cancer patients (Figure 1).

**Table 2 cells-12-01605-t002:** CD36 can be targeted using various antibodies and small molecules. All listed treatments are still in preclinical stages, except for VT1021, which has recently entered Phase II-III trials.

	Name of Compound	Disease Context	Description of Effect	References
Antibodies	FA6.152	Breast cancer, OSCC	Blocks all known functions of CD36, including interactions with TSP-1 and FA transporter properties. Anti-metastatic effects observed in vivo	[23,52]
	JC63.1	Breast, OSCC, Ovarian, Gastric cancers	Blocks FA and oxLDL uptake. Anti-metastatic effects observed in vivo	[14,23,52,54]
		Melanoma	Inhibits CD36 FA uptake on immunosuppressive T_regs_ and cytotoxic CD8^+^ T cells and restores anti-tumor immunity to TME	[56,73,81]
Peptides	Cyclic psap	Ovarian cancer	Induces expression of TSP-1 from MDSCs to promote apoptosis of cancer cells and endothelial cells	[75]
	EP80317	Cardiovascular diseases	Mitigates development of hypercholesterolemia and atherosclerosis. Also exhibits cardioprotective effects against myocardial ischemia and reperfusion injury	[109,110,111]
	VT1021	Breast Glioblastoma, Ovarian, Pancreatic cancers	Induces expression of TSP-1 from MDSCs to activate CD36- and CD47-mediated apoptotic signaling in cancer cells and endothelial cells. Also increases CTL infiltration as well as M1:M2 macrophage ratios	[92,93]
Small molecules	AP5258 and AP5156	Diabetes	Protects against diabetic atherosclerosis, dyslipidemia, and insulin resistance	[106]
	2-methylthio-1,4-naphthoquinone (MTN)	Glioblastoma	Blocks growth of glioblastoma CSCs	[16]
		Development	Anti-angiogenic properties	[88]
	Puerarin	Diabetes	Mitigates diabetic dyslipidemia by promoting FAO in skeletal muscle of diabetic rats	[107]
	Salvianolic acid (SAB)	Macrophages	Blocks macrophage uptake of oxLDL	[86]
		Obesity	Reduces visceral fat and improved insulin resistance in diet-induced obese mice	[87]
Synthetic FA analogs	FA-Linked Pt (IV)-Prodrugs	Ovarian cancer	Synthetic FA mimetic conjugated to cisplatin, selective drug uptake via elevated CD36	[74]
	Sulfo-N-succinimidyl oleate (SSO)	Breast, Cervical, Colorectal, Ovarian Cancers	Blocks uptake of long chain FAs and oxLDL	[17,52,53,61]
		Macrophages	Blocks uptake of long-chain FAs and oxLDL	[39,112]
		Diabetes	Corrects cardiomyopathy observed in diabetic rat hearts	[85]

## 9. Conclusions

In recent years, CD36 has quickly emerged as a promising therapeutic target to treat cancer, with broad clinical implications. CD36 expression has been shown to exhibit a prognostic value among patients across a wide spectrum of cancers, and pharmacological inhibition has demonstrated efficacy in numerous pre-clinical in vivo models. Many studies have implicated CD36-mediated FA uptake activity as its main mechanism of action in cancer cells (Table 1). Notably, CD36 overexpression in breast and gastric cancers has been demonstrated to mediate metabolic rewiring to favor FAO [17,40], as well as confer metabolic plasticity during nutrient deprivation [52]. Indeed, the previous work from our group demonstrated that under conditions of glucose deprivation, lapatinib-resistant breast cancer cells exhibit the induction of FAO to survive [52]. Over the previous decade, the importance of FA uptake has become increasingly recognized as a critical feature of cancer biology. It has long been thought that de novo lipogenesis is the predominant source of FAs for cancer cells [113]. However, recent metabolic labeling studies have demonstrated that exogenous uptake can supply >90% of intracellular FAs in fibroblasts and cancer cells (HeLa and H460) when palmitate is supplied at physiological conditions [114]. Interestingly, this study also demonstrated that, under these conditions, the cellular demand for FA synthesis via glycolysis-derived acetyl-CoA was reduced in cancer cells and that increased lipid scavenging was also associated with desensitization to the pharmacologic inhibition of glycolysis [114]. This finding was reminiscent of our previous observation that lapatinib-resistant breast cancer cells exhibiting the induction of CD36 were rendered less sensitive to glucose deprivation [52]. Moreover, we observed near stoichiometric conversion of glucose to lactate in these resistant cells, whereas sensitive parental cells demonstrated alternate metabolic fates, such as generating acetyl-CoA to feed the TCA cycle [52]. These studies suggest that CD36-mediated metabolic rewiring could have considerable implications in shifting cellular energetics away from classical Warburg anaerobic glycolysis to favor lipid metabolism in CD36-overexpressing tumors. However, the extent of CD36-mediated metabolic rewiring across cancer types or how this process contributes to tumor progression and drug resistance remains to be fully elucidated. In addition, cancer cells in acidic conditions have also been shown to favor the FAO of exogenously acquired FAs [115]. Interestingly, instead of feeding the TCA cycle, the predominant fate of acetyl-CoA generated from FAO in this study was determined to be mitochondrial protein hyperacetylation to restrain electron transport chain complex I activity to reduce ROS production [115]. While this study found no increase in the expression levels of CD36 or other FA transporters, it is tempting to speculate whether similar trends could be observed in cells overexpressing CD36, particularly in drug-relapsed tumors with acidic microenvironments. This could have immense implications in the context of drug resistance since the upregulation of antioxidant programs has been reported as being a critical mechanism in the establishment of drug-tolerant persister cells in many cancer types [116,117]. Whether CD36 facilitates this process remains a particularly important question.

We are only beginning to understand the complex role of CD36 as a mediator of the interaction between cancer cells and stromal cells in the TME. It is becoming clear that tumor–adipocyte crosstalk is indispensable for securing FAs to fuel the metabolically expensive process of cancer cell metastasis in some cancer types. CD36 is also implicated in the etiology of drug resistance mechanisms to a broad range of drug classes in cancer cells via both FA metabolism-dependent and -independent fashions, further adding complexity to the role of CD36 in cancer biology. While a lot has already been discovered about CD36 in other cell types, the role of CD36-mediated cell signaling in cancer cells remains largely uncharacterized and warrants further study. Importantly, CD36 has also been previously reported to lack glycosylation in several mouse and human cancer cell lines under basal culture conditions, indicating a probable defect in cell surface localization and FA uptake activity in these cells [15]. While several studies have reported that CD36 is expressed at the glycosylated ~88kDa size in cancer cells [14,15,17,20,40,51,53,57,61], it might be worth revisiting the glycosylation status and/or cellular localization of CD36 in future studies. In addition to its expression and activity in cancer cells, CD36 also has important implications in several intratumoral immune populations, where it likely contributes to treatment failure due to the development of an immunosuppressive TME. Thus, in addition to perturbing tumor growth, metastasis, and drug resistance via targeting CD36 in cancer cells, the concurrent inhibition of CD36 on intratumoral T_regs_, CD8^+^ T cells, and M2 TAMs would offer the additional bonus of eliciting anti-tumor immunity to further reduce the tumor burden (Figure 4). Lastly, given the fact that CD36 can be a therapeutic target of interest in the treatment of the common co-morbidities of cancer patients, such as atherosclerosis and diabetes, anti-CD36 therapy also has the potential to concurrently correct these systemic metabolic dysfunctions. In summary, the prospective benefits of CD36-targeted therapy are multifaceted and have the potential to be a paradigm-shifting approach in the battle against cancer.

## Figures and Tables

**Figure 3 cells-12-01605-f003:**
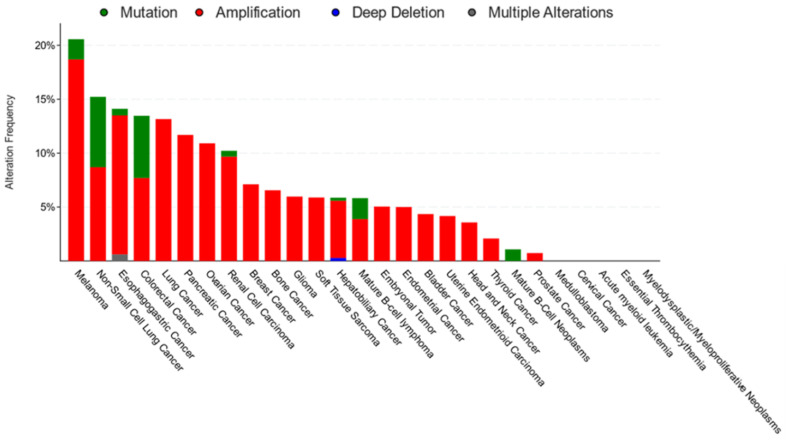
Genomic alterations of CD36 in cancer. CD36 was found to be mutated and/or exhibit copy number alteration in 8% of patients (202/2565) in a pan-cancer analysis [63]. The majority of these samples were from treatment naïve, primary cancers. CD36 amplification was the most common alteration observed across cancer types. CD36 mutations were also documented but have not been linked to carcinogenesis or drug resistance in ClinVar or COSMIC databases, respectively.

**Figure 4 cells-12-01605-f004:**
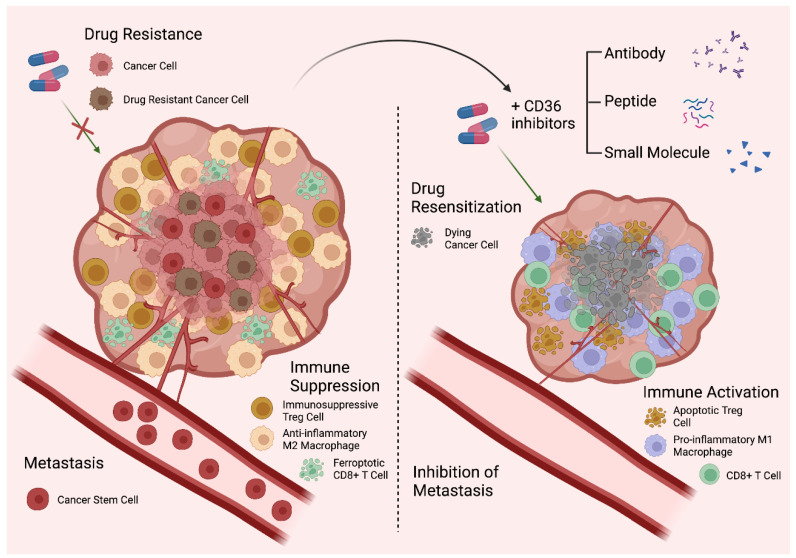
The multifaceted role of CD36 and its potential as a therapeutic target. CD36 promotes cancer progression not only by providing extracellular FAs to support the metabolic needs of cancer cells, but also by contributing to the maintenance of cancer cell stemness, facilitating metastasis, establishing immunosuppressive TMEs, and causing drug resistance (**left**). Thus, the inhibition of CD36 is expected to have a broad impact as a cancer treatment (**right**).

**Table 1 cells-12-01605-t001:** Roles of CD36 in cancer. CD36 can have metabolic, immunological, or cellular adhesion functions, depending on the cellular context. CD36 is expressed in a variety of different cell types in the tumor microenvironment, including mature cancer cells, CSCs, and various tumor-associated immune cells.

Cancer	Cell Type That Expresses CD36	Main Effects of CD36	References
Brain	CSCs	Stemness marker, promotes tumor initiating ability and self-renewal	[16]
Breast	Cancer Cells, CSCs	FA transporter, drug resistance, stemness marker, promotes EMT, metabolic reprogramming	[17,41,50,51,52]
	TAMs	Phagocytosis of apoptotic cells, recruitment to the TME, immune evasion	[66]
Cervical	Cancer Cells	FA transporter, tumor growth, metastasis, promoted EMT	[60,61]
Colorectal	Cancer Cells	FA transporter, metastasis	[53]
Gastric	Cancer Cells	FA transporter, metastasis, metabolic reprogramming	[20,40,54,55,67]
Leukemia	Cancer Cells, CSCs	FA transporter, drug resistance, metastasis	[58,68,69]
Liposarcoma	Cancer Cells	FA transporter	[50]
Liver	Cancer Cells, CSCs,	Stemness marker, promoted EMT, metabolic reprogramming	[59,70]
	TAMs	Immune evasion, metastasis	[71]
Melanoma	Cancer Cells	Drug resistance	[72]
	Intratumoral T_regs_, CD8^+^ T Cells	FA transporter, immune evasion	[56,73]
OSCC	CSCs	FA transporter, metastasis	[23]
Ovarian	Cancer Cells	TSP-1 receptor, FA transporter, drug resistance, metastasis	[14,74,75]
Pancreatic	Cancer Cells	Drug resistance	[57]
Prostate	Cancer Cells	FA transporter	[50,62,65]

## Data Availability

Data used to generate Figure 3 can be accessed at cbioportal.org, accessed on 20 February 2023.

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
