# Peer review of "The Role of CD36 in Cancer Progression and Its Value as a Therapeutic Target"

_cells, 2023, doi:10.3390/cells12121605_

Round 1
Reviewer 1 Report
The CD36 review article from Dr. Kurokawa lab is comprehensive and well written in general. A few minor point might help increase the impact.
1) Section 2 start line 69:
Regulation of CD36 expression should be strengthened. For example:
1. DOI: 10.7150/thno.34024
2. DOI: 10.3390/cells11233885
3. DOI: 10.1080/10715762.2022.2114904
2) line 258, It seems that CD36 regulates fatty acid oxidation in REF 20. Please double check.
3) It could be better to move the part of description of Ref 46 into section 6, line 285.
4) Line 388: Another approach to targeting CD36 would be to elicit the TSP-1-CD36-mediated 388
apoptotic signaling pathway … These papers could be referred to
Examples:
DOI: 10.1158/0008-5472.CAN-08-2940
DOI: 10.1038/71517
DOI: 10.1016/j.ygyno.2021.11.006

Reviewer 2 Report
In this review by Feng et al, the authors summarized the biological role of CD36 for metabolic fuel maintenance in the progression in several types of cancer and suggested its potential as a therapeutic target by antagonizing its function. This is a well written manuscript. However, a number of minor concerns need to be addressed before this manuscript is in a publishable fashion. Specific comments are as follows:
1) The trafficking of CD36 to the cell surface needs to be briefly introduced. As this protein is heavily N-glycosylated, does CD36 go through the secretory pathway? Is there a signal sequence structurally? How is the methionine got cleaved?
2) Are there potential ligands of CD36 to activate Stat3 and Sox2 in development of cancer stem cells and EMT (Figure 1B) for example fatty acids?
3) CD36 can be phosphorylated at extracellular residues by PKC and PKA. How does this occur? Are these residues phosphorylated before their surface expression?
4) The data from Figure 3 need to be better explained. For example, what does "CNA data" mean? What do the plus signs mean?
